

# Structural-topic aware deep neural networks for information cascade prediction

Bangzhu Zhou[1], Xiaodong Feng[2] and Hemin Feng[3]

[1] School of Management, University of Science and Technology of China, Hefei, China
[2] School of Information Management, Sun Yat-sen University, Guangzhou, China
[3] IMBA, The Chinese University of Hong Kong, Shenzhen, Shenzhen, China

## ABSTRACT

It is critical to accurately predict the future popularity of information cascades for many related applications, such as online opinion warning or academic influence evaluation. Despite many efforts devoted to developing effective prediction approaches, especially the recent presence of deep learning-based model, the structural information of the cascade network is ignored. Thus, to make use of the structural information in cascade prediction task, we propose a structural-topic aware deep neural networks (STDNN), which firstly learns the structure topic distribution of each node in the cascade, feeds it to a sequential neural network, and finally predicts the future popularity of the cascades. It can inherit the high interpretability of Hawkes process and possesses the high predictive power of deep learning methods, bridging the gap between prediction and understanding of information cascades by capturing indicative graph structures. We evaluate our model through quantitative experiments, where our model exhibits promising performance, efficiency higher than the baselines.

## INTRODUCTION

As of now, an escalating number of social media platforms, including platforms such as Facebook, Youtube, Sina Weibo, and others, have emerged, collectively placing the attention economy at the forefront of this era. The advent of these online social platforms has significantly reshaped the dynamics of information transmission among users, resulting in a substantial enhancement in the creation and dissemination of information. The process of information transmission between users can be conceptualized as an information cascade. Indeed, information cascades extend beyond social networks; the citation process of academic papers can similarly be construed as the formation of information cascades. Precisely forecasting the future extent of information cascades, indicating the anticipated popularity of specific online content, carries considerable significance. On the one hand, it proves advantageous in viral marketing, online advertising, and information recommendation; on the other hand, it may also give rise to adverse effects, such as the rapid dissemination of rumors.

Corresponding authors
Bangzhu Zhou,
zbz1480491537@mail.ustc.edu.cn
Xiaodong Feng,
fengxd5@mail.sysu.edu.cn

The prediction of information cascades relies on early dissemination characteristics to forecast their future reach. Nevertheless, due to the extensive and open nature of social platforms, coupled with external factors (*Cao et al., 2017*) such as network topology, follower relationships, user interests, posting times, and privacy considerations, the prediction task becomes inherently challenging. The vast scale of network users, dynamic information diffusion, rapid propagation speed, and the inherent stochastic nature of the pathways and processes involved contribute to the complexity. The uneven distribution of "popularity" among different pieces of information further intensifies the challenge. Consequently, accurately forecasting information cascades presents significant challenges within the context of these dynamic and intricate social systems.

In the prediction of information cascade, traditional methods for extracting features for popularity prediction can be broadly categorized into two groups: feature-based methods (*Szabo & Huberman, 2010*) and generative methods (*Li et al., 2017*). The feature-based approach involves extracting diverse features from information based on human prior domain knowledge, encompassing temporal features, structural features, content features, and others. Subsequently, regression/classification models are trained to predict the future popularity of the information. The challenge in this method lies in the selection of relevant features, with the quality of feature selection significantly impacting predictive performance. In contrast, generative methods aim to represent and model the process by which information attracts attention, facilitating a clearer understanding of the fundamental mechanisms governing information propagation dynamics. However, generative methods exhibit suboptimal predictive capabilities as they are not explicitly optimized for popularity forecasting.

In order to address the shortcomings of feature-based and generative methods in popularity prediction, scholars have begun to shift their focus towards deep learning methods. A model closely aligned with our research is the DeepHawkes model which introduced by *Cao et al. (2017)*. This model integrates the highly interpretable Hawkes model with the superior predictive accuracy of deep learning. DeepHawkes not only inherits the high interpretability of the Hawkes process but also possesses the strong predictive capabilities of deep learning methods, thus bridging the gap between predicting and understanding information cascades. However, it is crucial to note that the DeepHawkes model is tailored to model the information propagation process, neglecting the impact of network topology on propagation. In reality, network topology significantly influences information propagation, as different topological structures involve distinct users and propagation mechanisms. Different users engage with various topics, and these topics often exhibit varying degrees of popularity. Therefore, when predicting information propagation, the network topology emerges as an indispensable factor that should not be overlooked.

Fortunately, contemporary research in network science (*Bartlett & Cussens, 2017*; *Dai, Ren & Du, 2020*) has recently shifted its focus towards modeling the inherent topological structures within network architectures. A recent study (*Long et al., 2020*) has introduced an innovative approach to extract structural topics associated with each node in a graph. Drawing inspiration from Latent Dirichlet Allocation (LDA) and drawing an analogy

between document and graph data, this methodology integrates the acquired structural topics into a graph neural network framework for subsequent tasks, including node classification and link prediction. However, the effectiveness of automatically obtaining structural node representations for the cascade prediction task remains uncertain.

Hence, this study broadens its focus to encompass the structural aspects of cascade prediction tasks. It incorporates the structural topic representation assigned to each node into the deep neural network model, introducing a novel framework termed the Structural Topic-aware Deep Neural Network (STDNN) model. The STDNN model examines diverse network topologies, considers the self-excitation mechanism among these topologies, and integrates the influences of both cascading dynamics and network topology. In summary, the primary contributions of this article include:

- We propose an enhanced DeepHawkes model that incorporates structural themes. The Graph Anchor LDA (Latent Dirichlet Allocation) topic classification model is employed to extract structural topics from the network topology, and the deep model is utilized to learn the representation vectors of these structural topics, effectively simulating the self-excitation mechanism between them.
- The Structural Topic-aware Deep Neural Network (STDNN) model is introduced, which combines DeepHawkes, a deep neural network model that specifically solves the information cascade problem with the Graph structural-topic model. The integration ensures that the prediction model not only maintains the high interpretability of DeepHawkes but also thoroughly takes into account the characteristics of the network topology. It significantly enhances the model's ability to represent the information diffusion process, thereby improving the accuracy of popularity prediction.
- We conduct comprehensive experiments using real datasets to systematically compare the prediction results of multiple algorithms under different conditions to verify the effectiveness of the proposed method.

In the subsequent sections of this article, 'Related works' provides a comprehensive review of related literature, followed by 'Structural-topic aware deep neural network' which elaborates on the proposed methodology. 'Experiment setup' delineates the specific experimental configurations, with corresponding results presented in 'Experimental results'. Finally, the conclusions of this research are presented in 'Conclusions'.

## RELATED WORKS

In this section, we will mainly discuss the research most closely related to our work, including information popularity prediction and topic models.

The methods for predicting information cascades can be categorized into three distinct groups: feature-based methods, generative process methods, and deep learning-based methods. The feature-based methods usually regard the popularity prediction task as a regression problem (*Tsur & Rappoport, 2012*; *Cheng et al., 2014*) or a classification problem (*Szabo & Huberman, 2010*; *Romero, Tan & Ugander, 2013*; *Shulman, Sharma & Cosley, 2016*). The user features (*Cui et al., 2013*), content features (*Tsur & Rappoport, 2012*), structural features (*Zhang, Zeng & Tang, 2021*), and temporal features (*Pinto, Almeida*

*& Gonçalves, 2013*) for the cascades are meticulously crafted through manual design, drawing insights directly from the original data. These methods heavily rely on domain knowledge, making generalization challenging, and the manual feature extraction process is time-consuming and labor-intensive. Consequently, some scholars have turned their attention to generative processes for predicting information cascades.

Generative process methods aim to model the cascade propagation process, primarily relying on Poisson processes or Hawkes processes. For instance, concerning the future impact of academic papers, the reinforced Poisson process (RPP) (*Wang, Song & Barabási, 2013*) applied reinforcement mechanisms to model it. PETM (*Gao, Ma & Chen, 2015*) expanded upon the foundation of the RPP model through a temporal mapping process. *Feng et al. (2020)* proposed a feature-regularized reinforced Poisson process (FRRPP), which leveraged feature regression terms to capture the correlations between different posts. The second important method in generative approaches is the Hawkes process. SpikeM (*Matsubara et al., 2012*) integrated the merits of an epidemic model and the Hawkes model to simulate the actual transmission process of the cascade. The dual sentimental Hawkes process (*Ding et al., 2015*) considered combining self-excitation and cross-excitation mechanisms to model the impact of information. SEISMIC (self-exciting model of information cascades) (*Zhao et al., 2015*) utilized Hawkes processes to model temporal delay processes for predicting future retweet counts. A hybrid model composed of a Hawkes process has been proposed by *Mishra, Rizoiu & Xie (2016)*, which integrated the predictions of random forest with generative processes. However, these generative models primarily focus on the propagation process of information cascades and do not provide predictions for the future popularity of information.

In light of advancements in deep learning technology, numerous scholars have directed their research efforts towards establishing effective prediction models within the realm of deep learning. For example, *Li et al. (2017)* proposed the DeepCas model, which transformed the cascade graph as node sequences through random walk and learns the representation of each cascade under a deep learning framework. *Cao et al. (2017)* introduced DeepHawkes, an extension of DeepCas designed to address the limitations associated with neglecting the time decay effect. It utilized an end-to-end deep learning framework, drawing an analogy to the interpretable factors of the Hawkes process. DeepDiffuse (*Islam et al., 2018*) effectively captured the network among observed nodes, enabling accurate predictions of when the next node will participate in the cascade. The CasCN model (*Chen et al., 2019*), based on propagation paths, partitioned the cascade graph into multiple cascade subgraphs. It utilized an enhanced graph convolution approach to learn representations of cascade subgraphs, capturing the dynamic evolution of cascade networks. *Zhao, Zhang & Feng (2022)* presented CasTCN, which utilized the temporal and structural information of cascade networks as input to predict the future growth of information cascades. However, these methods either neglect the impact of cascade structure on information popularity prediction or excessively focus on subgraphs, resulting in poor model interpretability.

Another research area related to this study is topic models. Topic modeling is a widely used technique designed for text clustering, frequently employed to identify latent topic

information within extensive document sets or corpora. Latent Dirichlet Allocation (LDA) (*Blei, Ng & Jordan, 2003*) was the most typical topic models, which consisted of three layers of generative models. *Kou et al. (2018)* proposed the STTM model, which leveraged the spatial and temporal features of short texts in social networks to obtain more accurate semantics, thereby generating higher-quality topics. *Shi et al. (2019)* designed dynamic topic modeling *via* a self-aggregation method (SADTM). This method can capture aspects of the time-varying topic distribution and address issues related to sparsity. *Shi et al. (2020)* used information from users and followers, combining it with user topic models to uncover the search intentions and preferences of users in social networks. This effectively addressed issues related to semantic sparsity and historical data. However, these topic models primarily focus on the text features of social networks, often overlooking the topological structure patterns of the network. Consequently, *Long et al. (2020)* have introduced a Graph Structural-Topic Neural Network, who employed anonymous random walk to automatically learn structure characteristics of nodes in graphes. Nevertheless, this methodology has yet to be employed in predicting issues related to information cascades.

Based on the aforementioned, this article introduces a novel information cascade prediction method named STDNN. It comprehensively integrates the high interpretability and predictive performance of DeepHawkes with the capability of the Graph Structural-Topic Neural Network to capture the topological structure of cascade networks.

## STRUCTURAL-TOPIC AWARE DEEP NEURAL NETWORK

In this section, we introduce our model STDNN. Figure 1 gives an overview of our model STDNN and Fig. 2 depicts the structure topic learning process. The backbone of our STDNN model primarily consists of Graph Anchor LDA and recurrent neural network (RNN), divided into three main components, as shown in Fig. 1. The first phase is the user embedding process, where we utilize topic learning based on the Graph Anchor LDA model to comprehensively consider the influence of network topology on user embedding. The second section focuses on subsequent path encoding and pooling. Specifically, it entails feeding the forwarding paths, which contain network structures, into a recurrent neural network (RNN). Subsequently, the values of the last hidden layer are summed and pooled, facilitating the modeling of the self-excitation mechanism within the forwarding paths. The final component involves the non-parametric time decay effect, which models the temporal decay process of information propagation using non-parametric methods. Notably, we have considered the influence of network topology on information cascade popularity. We commence by providing foundational definitions and introducing the methodology for extracting structural topics.

### Preliminaries

**Definition 1.** *Information Cascades.* Suppose we have $M$ messages, denoted by $M = \{m^i\}(1 \leqslant i \leqslant M)$. For each message $m^i$, it is denoted by a cascade $C^i = \left\{ \left( u_j^i, v_j^i, t_j^i \right) \right\}$ to record the diffusion process of it, where the tuple $(u_j^i, v_j^i, t_j^i)$ corresponds to the $j$th retweet, meaning that user $v_j^i$ retweets message $m^i$ from user $u_j^i$, and the time elapsed between the

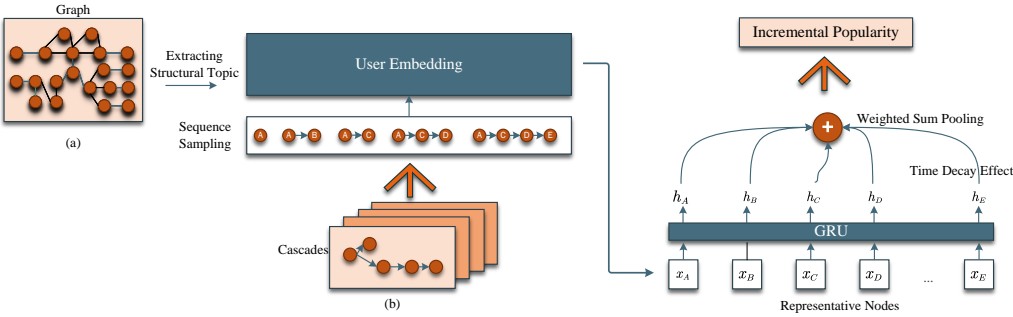

**Figure 1** **An overview of structural-topic aware deep neural networks (STDNN).** The model consists of two major components: (A) Extracting structural topic of graph, (B) Deep neural networks.

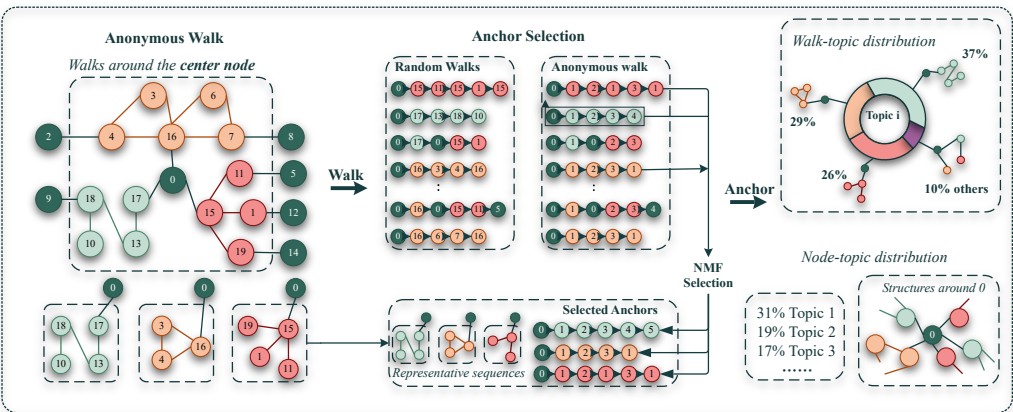

**Figure 2** **Framework of extracting structural topic of graph anchor LDA model.**

original post and the $j$th retweet is $t_j^i$. The popularity $R_t^i$ of message $m^i$ up to time $t$ is defined as the number of its retweets, *i.e.,* $|\{(u_j^i, v_j^i, t_j^i)|t_j^i \leqslant t\}|$.

The representation of the topological structure of a cascade within a diffusion network is referred to as a cascade graph. This cascade graph can correspond to one or more pieces of information, necessitating manual extraction. For instance, an information cascade sample may correspond to a topological graph representation. Additionally, datasets may directly provide the topological structure of the cascade graph. This type of topological graph is typically considered a global network, encompassing the topological structural relationships of all information cascade sequences in the entire dataset.

**Definition 2.** Incremental Popularity Prediction. Given the forwarding path $\{(u_j, v_j, t_j)|t_j \leqslant T\}$ of cascade $C$ within the period $[0, T)$ (commonly referred to as the observation window), the corresponding cascade graph is denoted as $G^T = (V^T, E^T)$. The task of predicting cascade increment popularity involves a regression problem, aiming to predict the increment in the number of engaged nodes in cascade $C$ after a time interval $\Delta t$, denoted as $\Delta s_C = |V_C^{T+\Delta t}| - |V_C^T|$. A specific instance of this prediction task is the final

popularity prediction, which anticipates the increment $\Delta s_C^\infty = |V_C^\infty| - |V_C^T|$ between the number of observed nodes and the number of final nodes in the cascade.

## Extracting structural topic of nodes

In the realm of natural language processing, topic models play a crucial role in elucidating the distributional distinctions embedded in higher-order structural patterns. This study undertakes a theoretical analysis to scrutinize the underlying principles of acquiring substructure topic distributions within graph networks. Subsequently, an adaptive graph neural network is introduced to adeptly leverage such structural information. Please refer to Fig. 2 for an intuitive illustration. In particular, we extract structural patterns through anonymous walks (*Micali & Zhu, 2016*). Anonymous random walks are sampled for each node to characterize the local structures of a node. Anonymous walks involve obscuring the true identity of nodes during a random walk, retaining only the serial number ID as an identifier to document the transitional rules within the walk. For each node, a set of random walk sequences with a predefined length is sampled. Subsequently, their potential distributions of anonymous walk experiences, along with the overall average experience distribution across the entire graph, are calculated to construct the authentic distribution.

We argue that anonymous walk does not share the identity space of nodes, that is, the id of nodes in each walk will be counted from scratch, so that the id of nodes in multiple walks will be repeated. The sequence generated by a random walk, as depicted in the figure, is transformed into the sequence generated by an anonymous walk. For instance, $(0, 15, 11, 15, 1, 15)$ becomes $(0, 1, 2, 1, 3, 1)$ in Fig. 2. *Long et al. (2020)* introduced an algorithm for selecting an anchor structure ('Anchor'), aiming to filter representative structural features of the network, thereby reducing representation complexity and mitigating noise interference. We refer to their practice in the follow-up processing.

Given a graph $G = (V, E)$, by anonymous walk, two matrices can be learned to depict the structural information of the nodes in graph. One is node-topic matrix $N \in \mathbb{R}^{|V| \times Q}$, where each row $N_i$ corresponds to a distribution and $N_i^q$ denotes the probability of node $v_i$ belonging to the $q$-th structural topic. The second is walk-topic matrix $M \in \mathbb{R}^{Q \times |A_l|}$, where each row $M_q$ represents the topic distribution over walks in $A_l$ and $M_q^a$ denotes the probability of walk $a \in A_l$ belonging to the $q$-th structural topic. Here, $A_l$ signifies a set of possible anonymous walks with a length of $l$, and $Q$ represents the number of desired structural topics.represents the desired number of structural topics. Additionally, we define the set of anonymous walks starting from $v_i$ as $B_i$. According to the definition of Graph Anchor LDA (*Long et al., 2020*), we need to get the node-topic matrix $N$, meaning of getting the distribution matrix of each node topic. We first identify the 'anchor' and then proceed with topic modeling. Specifically, we set the walk-walk co-occurrence matrix $W \in \mathbb{R}^{|A_l| \times |A_l|}$, with $W_{i,j} = \sum_{v_q \in V} \mathbb{I}(w_i \in B_q, w_j \in B_q)$, and utilize non-negative matrix factorization (NMF) to extract anchors

$$H, Z = \arg \min \|W - HZ\|_F^2$$

s.t. $\quad H, Z^T \in \mathbb{R}^{|A_l| \times \alpha}, H, Z \geq 0.$ \hfill (1)

**Table 1  Key symbols of STDNN.**

| Symbol | Description |
| --- | --- |
| $m^i$ | Label for text content of the message |
| $C^i$ | Label for diffusion process of a message cascade |
| $u_j^i$ | The u represents the user who was forwarded, i represents that the i-th message was forwarded, and j represents that the current forwarding is the j-th forwarding |
| $v_j^i$ | The v represents represents the forwarding user, i represents that the i-th message was forwarded, and j represents that the current forwarding is the j-th forwarding |
| $t_j^i$ | The t represents the time interval between the moment when user v retweets a message and the moment when user u retweets the same message. |
| $R_t^i$ | Label for the actual number of messages forwarded |
| $\Delta s_C$ | Label for forwarding increment within $\Delta t$ time |
| $\Delta s_C^\infty$ | Label for total forwarding increment |
| $N, S, M$ | Labels for node-topic matrix, node-walk matrix and walk-topic matrix |
| $Q, l$ | Labels for the number of topics and the length of walks |
| $A_l$ | Label for a set of possible anonymous walks with a length of $l$ |
| $B_i$ | Label for the set of anonymous walks starting from node $v_i$ |
| $W$ | Label for the walk-walk co-occurrence matrix |

We iteratively update $H, Z$ until convergence, and then identify the anchors using $\mathbf{A}_q = \arg\ \max(Z_q), q = 1, \ldots \alpha$, where $A$ is the set of indices for anchors, and $Z_q$ is the $q$-th row of $Z$. Based on the selected anchors, we can learn the walk-topic distribution $M$. In addition, we define node-walk matrix as $S \in \mathbb{R}^{|V| \times |A_l|}$ with $S_i^a$ denoting the occurrences of $a$ in $B_i$. We finally get the node-topic distribution $N$ through $N = SM^\dagger$, where $M^\dagger$ denotes pseudo-inverse. Table 1 summarizes the symbols and their corresponding meanings of our model.

## User embedding

Each user in the generated retweet path is denoted as a one-hot vector, $h \in \mathbb{R}^{|V|}$, where $|V|$ is the total number of users. All users share an embedding matrix $N \in \mathbb{R}^{|V| \times Q}$, where $Q$ is an adjustable dimension of embedding, $N$ is equal the node-topic matric. This user embedding matrix converts each user into its representation vector:

$$x = N^T h, \tag{2}$$

where $x \in \mathbb{Q}^K$ and $N^T$ is the transpose of $N$. It is worth noting that the user embedding matrix $N$ is learned from the overall structure of the graph, which can represent the whole information of the graph to some extent. Therefore, the learned user embeddings are optimized for the initial supervised framework.

## Topic path coding

The influence among topics (influence transfer) and the significance of topics in topic path structure can be modeled through the GRU in recurrent neural networks

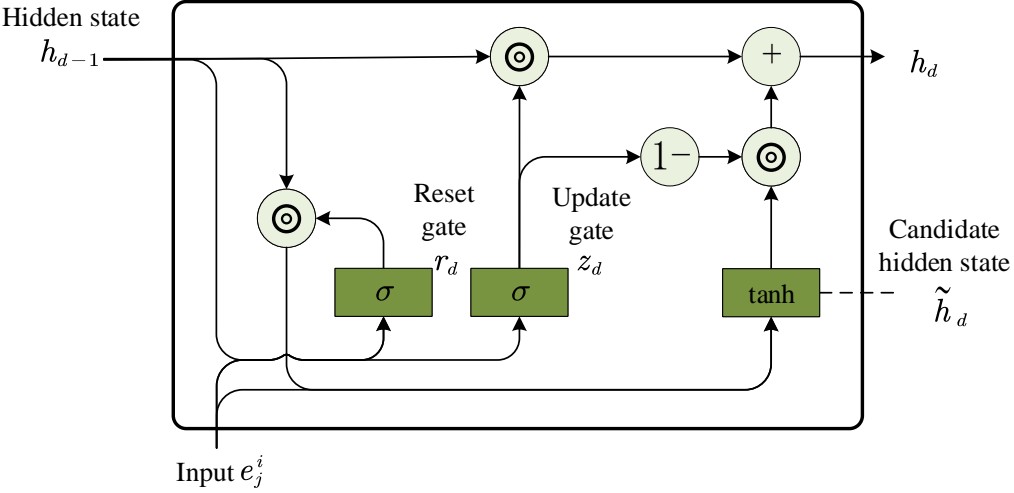

**Figure 3   GRU model.**

(*Mikolov et al., 2010*). The influence transfer from one prominent subject to another influences the prominence of the latter. The importance of topics in the topic path structure is determined by their frequent occurrence across multiple topic paths. Each theme path has the most GRU output pooled layer, which achieves Hawkes' self-exciting mechanism by accumulating various effects.

When GRU is used to encode each topic $e_j^i \left(1 \leqslant j \leqslant Q\right)$ of message $m^i$, where $e_j^i$ is the element of matrix $x$, the $d$-th hidden state $h_d = GRU\left(e_j^i, h_{d-1}\right)$ in GRU, where $h_d \in \mathbb{R}^{H'}$ is the output, and the topic represents the vector when $e_j^i \in \mathbb{R}^Q$ is input. $h_{d-1} \in \mathbb{R}^{H'}$ is the previous hidden state, $Q$ is the dimension of topic embedding, and $H'$ is the dimension of hidden state. The GRU model is shown below in Fig. 3. Firstly, the reset gate $r_d \in \mathbb{R}^{H'}$ is computed by

$$r_d = \sigma\left(W^r e_j^i + Y^r h_{d-1} + b^r\right), \tag{3}$$

where $\sigma$ is the sigmoid activation function, $W^r \in \mathbb{R}^{H' \times Q}$, $Y^r \in \mathbb{R}^{H' \times H'}$ and $b^r \in \mathbb{R}^{H'}$ are GRU parameters learned during training.

Secondly, the update gate $z_d \in \mathbb{R}^{H'}$ is computed by

$$z_d = \sigma\left(W^z e_j^i + Y^z h_{d-1} + b^z\right), \tag{4}$$

where $W^z \in \mathbb{R}^{H' \times Q}, Y^z \in \mathbb{R}^{H' \times H'}$ and $b^z \in \mathbb{R}^{H'}$.

Then, the actual activation of hidden state $h_d$ is computed by

$$h_d = z_d \odot h_{d-1} + (1 - z_d) \odot \tilde{h}_d, \tag{5}$$

where

$$\tilde{h}_d = \tanh\left(W^h e_j^i + r_d \odot \left(Y^d h_{d-1}\right) + b^d\right). \tag{6}$$

$\odot$ represents element-wise product, $W^d \in \mathbb{R}^{H' \times Q}, Y^d \in \mathbb{R}^{H' \times H'}$ and $b^d \in \mathbb{R}^{H'}$.

## Modeling the time decay effect

Given the temporal decay of retweet effects, we incorporate the time decay effect through a non-parametric approach. Let us consider the propagation of all messages within a time interval $T$, and assume the unknown practical delay effect is a continuously changing function within the range $[0, T)$. We partition the time length T into P intersecting intervals $\{[t_0 = 0, t_1), [t_1, t_2), \dots, [t_{P-1}, t_P = T)\}$ to estimate this time delay function and derive the corresponding discrete time delay variable $\lambda_p$. Assuming $t_j^i$ represents the time elapsed from the original post to the $j$th retweet of message $m^i$, and $g(T - t_j^i)$ denotes the corresponding time interval for the time decay effect of $j$th retweet, then the mapping function $g$ from continuous time to discontinuous time is defined as:

$$g\left(T - t_j^i\right) = p, \text{if } t_{p-1} \leqslant T - t_j^i < t_p. \tag{7}$$

Based on the time decay effect, the cascade topic $c_e^i$ of $m^i$ message can be denoted as:

$$c^i = \sum_{j=1}^{G_T^i} \lambda_{g\left(T - t_j^i\right)} h_j^i \tag{8}$$

where $c^i \in \mathbb{R}^{H'}$ represents the final representation of cascade $C$, which is assembled by the sum pooling mechanism, and for each retweet path $e_j^i$, we use the last hidden states as the representation of the entire diffusion path, denoted as $h_j^i$.

## Popularity prediction

Finally, the output from the sum pooling layer ($c^i$) is conveyed as input to the fully connected layer, known as the multi-layer perceptron (MLP). The MLP constitutes a feedforward artificial neural network model. We employ the representation vector from the sum pooling layer, as introduced in our model, directly as the input for the MLP, with the resulting output serving as the predicted cascade growth scale. This can validate the effectiveness of our sum pooling layer representation vector after topic learning, path encoding and time delay effect processing through the implementation of a straightforward model. The cascade representation vector derived from the pooling layer ($c^i$) can be directly fed into the MLP model, culminating in the retrieval of information cascade popularity prediction outcomes:

$$\hat{y}_t^i = MLP\left(c^i\right). \tag{9}$$

The optimization objective function is defined as:

$$Obj_{\min} = \frac{1}{m} \sum_{i=1}^{m} (\log \hat{y}_t^i - \log y_t^i)^2, \tag{10}$$

where $\hat{y}_t^i$ is the predicted incremental popularity of cascade $C$, $y_t^i$ is the real incremental popularity and $m$ is the total number of messages.

## EXPERIMENT SETUP

In this section, we put forward a comprehensive empirical experiment to evaluate the effectiveness of our model (STDNN) and compare the prediction performance of our model with state-of-the-art approaches.

### Data sets

We evaluate the performance of proposed model on two scenarios of information diffusion popularity prediction and compare with state-of-the-art methods to verify the effectiveness and generality of our model. The first scenario is to predict the future size of re-tweet cascades on Sina Weibo, and the second one is to forecast the citation count of papers from American Physical Society (APS). The weibo data is available at Github (https://github.com/CaoQi92/DeepHawkes) (*Cao et al., 2017*). The APS data comes from public datasets (https://journals.aps.org/datasets/inquiry). All requests will be sent to APS staff for review, and then researchers will receive a copy of this request *via* email. The dataset we used is available at Zenodo (https://zenodo.org/badge/latestdoi/670429356).

In the first scenario, the dataset is sourced from Sina Weibo, a social media platform based on user relationships and one of the most popular microblogging platforms in China. To ensure data fitting, we reorganize the Weibo dataset provided in *Cao et al. (2017)*, which captures all original tweets generated on June 1st, 2016, and tracks all retweets of each message within the subsequent 24 h. In Fig. 4A, the distribution of cascade popularity, representing the number of retweets for each message, follows a power-law distribution. Notably, in contrast to the experimental setup in DeepHawkes (*Cao et al., 2017*), the observation time window in this study is limited to only 1 h. After data arrangement and selection, we obtain 42,183 cascade graphs, comprising 1,390,020 nodes, each corresponding to a unique Weibo user. Subsequently, a social network graph with information on all node connections is constructed based on the selected graphs. Next, the dataset's 29,529 cascade graphs are allocated to the training set, 6,328 to the validation set, and 6,327 to the test set.

The second dataset is derived from the American Physical Society (APS), one of the most prestigious professional physics societies globally, with a history spanning 122 years since its establishment. This dataset encompasses all papers published by the 11 APS journals from 1893 to 2009, along with the citations among these papers. In this context, the number of days elapsed since the publication of the cited paper is recorded. The dataset comprises 24,338 cascade graphs, containing 122,975 nodes. A cascade consists of all the citations to a paper, and the number of citations reflects the cascade's popularity. Figure 4B illustrates the distribution of cascade popularity. Subsequently, we partition the dataset into training, validation, and test sets, comprising 17,037, 3,651, and 3,650 cascades, respectively.

In summary, the Weibo dataset comprises 42,183 information cascades, while the APS dataset contains 24,338 information cascades. In both scenarios, nodes represent users or papers, and each sample in the dataset constitutes an information cascade graph documenting the diffusion of a target message or paper. Additionally, we allocate 70% of

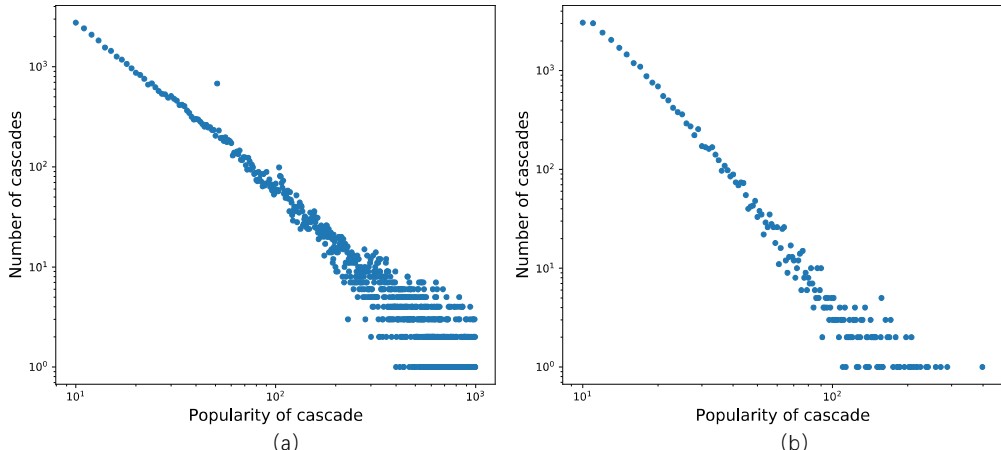

**Figure 4** **Distribution of popularity.** (A) The Sina Weibo dataset; (B) the APS dataset.

the cascades to the training set, the middle 15% to the validation set, and the remaining 15% to the test set.

## Evaluation metric

In order to evaluate the accuracy of predictions, following the practice of related models, we use the mean square log-transformed error (MSLE) and the median square log-transformed error (mSLE). Denote $M$ the total number of messages, and $SLE_i$ the square log-transformed error for a given message $m_i$, MSLE is employed to measure the error between predicted values and actual values, and it is defined as:

$$MSLE = \frac{1}{M} \sum_{i=1}^{M} SLE_i \qquad (11)$$

The $mSLE$ is capable of effectively mitigating the impact of outliers, which is defined as:

$$mSLE = median(MSLE_i) \qquad (12)$$

where $SLE_i = \left(\log \hat{y}_i - \log y_i\right)^2$, $\hat{y}_i$ is the predicted increment of the popularity for message $m_i$ and $y_i$ is the actual increment of the popularity.

## Baseline methods

In this section, we have chosen six state-of-the-art methods for comparative analysis with our proposed approach, namely Feature-linear/deep, DeepCas, DeepHawkes, CasCN, and CasTCN, respectively. It is worth noting that the models under consideration employ comparable or identical datasets as inputs for a fair comparison. Subsequently, we provide a detailed description of these baseline models.

### *Feature-based approaches*

Feature-linear and Feature-deep (*Cheng et al., 2014*) are feature-based methods to implement information cascade prediction. For Feature-linear, temporal features and

structural features of the cascade graph are input into L2 regularized regression for prediction. To establish a more robust baseline, the hand-crafted features are also fed into an MLP for forecasting the future size of the cascade, referred to as Feature-deep.

### A diffusion model-based approach

LIS (*Wang et al., 2015*) is a method that simulates cascade dynamics by learning two low-dimensional potential vectors from observed cascade information. These vectors are designed to capture the influence and sensitivity of the cascades, respectively.

### Deep learning-based approaches

- DeeCas (*Li et al., 2017*) is an end-to-end neural graph network framework based on GRU, attention mechanics and MLP to predict the size of the cascade prediction.
- DeepHawkes (*Cao et al., 2017*) is a model that depicts the factors of key mechanisms in the generative process and has a good understanding of the propagation process of messages.
- CasCN (*Chen et al., 2019*) is a framework based on graph convolutional networks (GCN) designed to capture both temporal and structural information for cascade prediction. The approach involves sampling a sequence of sub-cascade networks from a larger cascade network and subsequently employs graph convolutions to learn the representations of each sub-cascade.
- CasTCN (*Zhao, Zhang & Feng, 2022*) proposes a network-level rather than node-level deep neural network-based information cascade predictor, which extracts sub-cascade networks at distinct time intervals. This model employs a dynamic mapping mechanism on these sub-cascade networks to derive corresponding degree distribution sequences, and subsequently inputs them into a temporal convolutional network to effectively model time-dependent information.

## Parameter setting

For the baselines above and our proposed structural-topic aware deep neural networks model (STDNN), in order to obtain the best results for each data validation set, we adjust the hyper-parameters. For instance, the L2-coefficient is chosen from $\{10^{-8}, 10^{-7}, \ldots, 0.01, 0.1\}$ in feature-linear. For LIS, the parameter 231 setting consist with *Wang et al. (2015)*. For LIS, the parameter setting consist with *Wang et al. (2015)*. For the MLP, the dimensions of each layer are $\{512, 256, 128, 64, 32\}$. Sigmoid functions are employed as activation functions in the hidden layers, while the output layer utilizes [mention the specific activation function]. The learning rate is set at 0.01. For all the deep learning-based approaches, we follow the setting of DeepCas, where the embedding size of users is set to 50, and for DeepCas, DeepHawkes, and CasCN, the GRU in the hidden layer is configured with 32 units. The hidden dimensions of the two-layer fully connected layers in all MLP-based predictors are set to 32 and 16, respectively. The CasTCN, all parameters involved are consistent with those in the original paper. In deep neural networks, we set the time interval of non-parametric time decay effect to 10 min for Sina Weibo and 3 months for APS.

**Table 2  Overall prediction performance.**

| Datasets | Weibo dataset | | APS dataset | |
|---|---|---|---|---|
| Metric | MSLE | mSLE | MSLE | mSLE |
| Features-deep | 3.450 | 1.084 | 2.195 | 0.845 |
| Features-linear | 3.282 | 0.971 | 1.752 | 0.796 |
| LIS | 3.497 | 1.476 | 2.486 | 0.938 |
| DeepCas | 2.773 | 0.868 | 1.804 | 0.732 |
| DeepHawkes | 2.604 | 0.816 | 1.363 | 0.688 |
| CasCN | 2.593 | 0.798 | 1.348 | 0.652 |
| CasTCN | 2.581 | 0.746 | 1.319 | 0.633 |
| STDNN | **2.530** | **0.679** | **1.250** | **0.607** |

**Notes.**
Results for the proposed model are shown in bold.

# EXPERIMENTAL RESULTS

## Overall performance

The prediction performance of our proposed STDNN and the state-of-the-art baselines on both the Weibo Dataset and the APS Dataset is demonstrated in Table 2. Intuitively, our proposed model demonstrates superior performance compared to all baselines in predicting information cascades for the two scenarios, as assessed by the MSLE and mSLE evaluation metrics. Furthermore, our model significantly outperforms feature-based and diffusion model-based approaches, *e.g.*, Features-deep, LIS. It also outstrips the state-of-the-art deep learning approaches, *e.g.*, DeepCas, CasCN and CasTCN. Now, in the following, we compare the differences and effectiveness of the MSLE-based metric among our proposed model and these baselines and analyze the reasons in detail.

It is observed that features-deep does not outperform features-linear on both the Weibo and APS datasets. This indicates that, when a set of appropriate features is provided, the linear method is not necessarily inferior to the deep learning approach. However, the error of features-linear results is significantly higher than the error of our proposed model predictions. This emphasizes our previous claim that feature-based approaches heavily rely on manually crafted features, rendering them challenging to generalize across diverse scenarios.

For the diffusion model-based methods, LIS did not achieve satisfactory results in predicting information cascades; it performed the poorest among all methods. The main reason is that diffusion model-based methods like LIS typically model the propagation process of information cascades but do not effectively predict the future popularity of information cascades. Therefore, LIS performed the worst on both datasets.

For the deep-learning approaches, STDNN has also performed better than state-of-the-art baseline methods. DeepCas relies solely on random walk strategies, failing to capture crucial information about the network structure, leading to subpar performance in cascade prediction tasks. On the other hand, DeepHawkes, encoding the dynamics of cascades using Hawkes processes, combines the advantages of generative processes and deep learning, resulting in better performance compared to DeepCas. However, DeepHawkes does not

extract structural information from cascade networks, leading to inferior performance compared to the latest models like CasCN and CasTCN.

Our proposed STDNN outperforms all peers on all datasets and surpasses the state-of-the-art methods in information cascade prediction. Regarding the reasons, firstly, our model rigorously incorporates the topological structure of individual nodes within the network, facilitating a more precise and comprehensive modeling of the entire cascade network when contrasted with approaches reliant on subgraph modeling. Secondly, the utilization of Hawkes processes to encode the propagation of the entire cascade enhances the interpretability of the model. Our approach that not only combines the advantages of deep learning and generative methods but also takes into account the network topology of each cascade, results in highly satisfactory performance.

## Ablation experiments

To better analyze the impact of different factors on cascade prediction in the STDNN model, we design several variants of the STDNN:

- **STDNN-Linear**: In STDNN-Linear, we refrain from optimizing the user embedding process. In this configuration, our model effectively degenerates to be consistent with the DeepHawkes model. We directly utilize the representation vectors learned from the data as user embeddings.
- **STDNN-Node**: In STDNN-Node, we do not take the original node's thematic features as user embeddings. Instead, we employ the subgraph of node distributions as input, specifically using the node-walk matrix as user embeddings.
- **STDNN-Path**: In STDNN-Path, we focus solely on the influence of each forwarding event in the Hawkes process, rather than encoding the impact of the entire structural topic path through the GRU structure.

The specific performance of these variant models on the two datasets is summarized in Table 3 and illustrated in Fig. 5. Firstly, STDNN-Linear demonstrates a significant advantage over traditional feature-based methods and diffusion-based methods, as our model aligns with DeepHawkes in this configuration. Secondly, for STDNN-Node, using the walk-topic matrix as user embeddings yields better performance than STDNN-Linear, indicating the crucial role of network topology in the prediction process. However, its performance is inferior to our overall STDNN model, implying that considering the network topology of individual nodes yields better predictive performance than focusing solely on subgraph structures. Finally, in STDNN-Path, considering only the influence of single forwarding events significantly reduces the prediction accuracy, which indicates that future popularity is not only influenced by the current forwarding user but also by the entire forwarding path. This further emphasizes the necessity of the proposed approach.

## Parameter sensitivity

We primarily conduct sensitivity analyses on two parameters in our model, specifically, the length of walks $l$ and the number of topics $Q$. Following related work, we employ the MSLE as a metric to assess sensitivity to parameters on two datasets. For the length of walks $l$, as

**Table 3  Prediction performance of variants of the STDNN.**

| Datasets | Weibo dataset | | APS dataset | |
|---|---|---|---|---|
| Metric | MSLE | mSLE | MSLE | mSLE |
| STDNN-linear | 2.604 | 0.816 | 1.363 | 0.688 |
| STDNN-Node | 2.593 | 0.763 | 1.325 | 0.673 |
| STDNN-Path | 2.977 | 0.874 | 1.974 | 0.752 |
| STDNN | **2.530** | **0.679** | **1.250** | **0.607** |

**Notes.**
The best results are shown in bold.

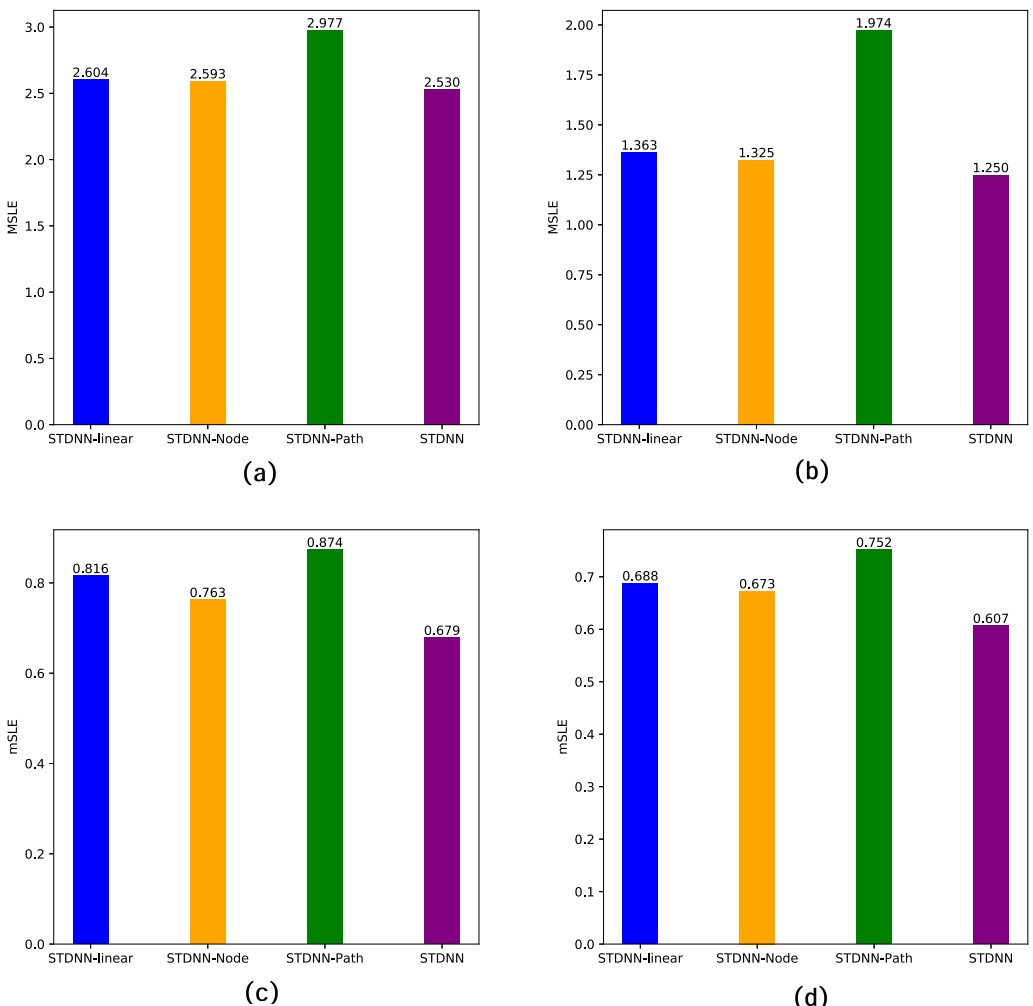

**Figure 5  Ablation study of STDNN on two data sets.** Where (A) MSLE of Weibo; (B) MSLE of APS; (C) mSLE of Weibo; (D) mSLE of APS.

depicted in the Fig. 6, there is a noticeable "V"-shaped variation pattern in MSLE with the increase of $l$ on both datasets. It's crucial to avoid setting values that are excessively large or too small, as they can adversely impact the model's performance. However, the fluctuation

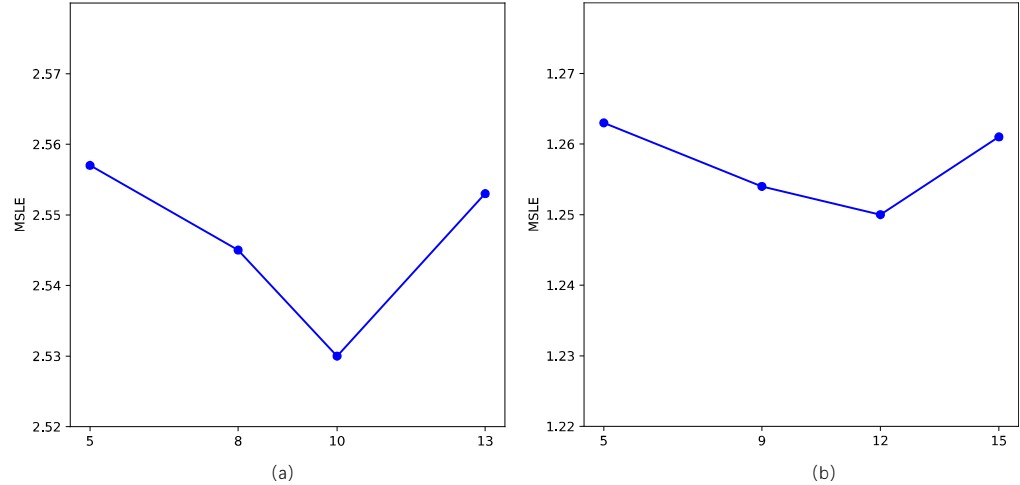

**Figure 6  Impact of the length of walks (l) on STDNN.** (A) The result of Weibo; (B) the result of APS.

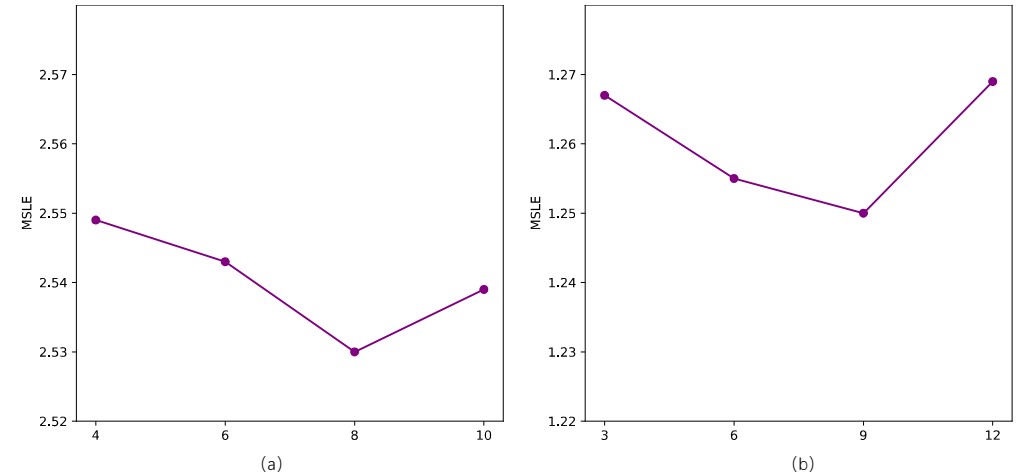

**Figure 7  Impact of the number of topics (Q) on STDNN.** (A) The result of Weibo; (B) the result of APS.

range is minimal, with a slight difference between the maximum and minimum values of MSLE, namely 2.557 *vs* 2.530 and 1.263 *vs* 1.250, respectively. Concerning the number of topics $Q$, as shown in Fig. 7, with the number of topics $Q$ increases, the performance on the two datasets exhibits a slight fluctuation in MSLE, specifically ranging from 2.53 to 2.55 and from 1.25 to 1.27, respectively. These experimental results indicate that our model is not sensitive to the parameters of the length of walks $l$ and the number of topics $Q$, making it relatively easy to implement in practice.

## CONCLUSIONS

In this article, we extend the DeepHawkes model by considering the impact of cascade topology on diffusion dynamics, and propose the STDNN model, which integrates the Graph Anchor LDA topic model into the DeepHawkes framework. This model not only combines the benefits of deep learning and generative methods but also integrates the Graph Anchor LDA model to extract structural information from nodes, which leads to a more comprehensive representation of the information diffusion process. The STDNN model initiates by extracting structural topic information from nodes, optimizing the user embedding process to encapsulate the network topology of nodes. Following this, it employs the Hawkes process to encode cascade propagation paths, encompassing both the self-excitation mechanism during forwarding and the temporal delay effects during propagation. This holistic approach contributes to the ultimate enhancement of predictive performance. STNDD consistently exhibits superior performance over other baseline methods when evaluated on two publicly available datasets, Sina Weibo and APS.

In our future work, we will try to conduct a more in-depth exploration of information pertaining to cascading nodes, including temporal attributes associated with nodes. Furthermore, we intend to investigate the amalgamation of STDNN with other deep neural network architectures, seeking to identify more effective predictive methodologies.

### Funding

The authors received no funding for this work.

### Competing Interests

The authors declare there are no competing interests.

### Author Contributions

- Bangzhu Zhou conceived and designed the experiments, performed the experiments, analyzed the data, performed the computation work, prepared figures and/or tables, authored or reviewed drafts of the article, and approved the final draft.
- Xiaodong Feng conceived and designed the experiments, performed the experiments, analyzed the data, performed the computation work, prepared figures and/or tables, authored or reviewed drafts of the article, and approved the final draft.
- Hemin Feng conceived and designed the experiments, performed the experiments, analyzed the data, performed the computation work, prepared figures and/or tables, authored or reviewed drafts of the article, and approved the final draft.

### Data Availability

The APS data can be requested at https://journals.aps.org/datasets/inquiry. All requests will be sent to APS staff for review, and then researchers will receive a copy of this request *via* email.

The dataset we used is available in the Supplemental File and at Zenodo: zbz1480491537. (2023). zbz1480491537/supreme-tribble: The original data for the paper Structural-topic aware deep neural networks for information cascade prediction (v1.0.0). Zenodo. https://doi.org/10.5281/zenodo.8180848

The weibo data is available in the Supplemental File and at GitHub: https://github.com/CaoQi92/DeepHawkes) which links the data to a Shared Network Disk (https://pan.baidu.com/s/1c2rnvJq; password: ijp6.)

The raw measurements and all the related codes are available in the Supplemental Files.

## Supplemental Information

Supplemental information for this article can be found online at http://dx.doi.org/10.7717/peerj-cs.1870#supplemental-information.

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
