# Peer review of "Structural-topic aware deep neural networks for information cascade prediction"

_PeerJ Computer Science, doi:10.7717/peerj-cs.1870_

## Round 0.1 · original submission · Major Revisions

Based on the comments of the reviewers, the work is interesting. However, it needs further revision in writing, experiments, etc. Please refer to the comments and revise the article accordingly.

Reviewer 3 has requested that you cite specific references. You may add them if you believe they are especially relevant. If you do not include them, this will not influence my decision.

Reviewer 1 ·

Basic reporting

NA

Experimental design

NA

Validity of the findings

NA

Additional comments

NA

Cite this review as

Reviewer 2 ·

Basic reporting

This paper propse a new method which is structural-topic aware and based on deep neural network. It's ttheoretical basis is reasonable with relative clear definitions of all terms and detailed proofs. The downside is that the description of STDNN is not sufficient: which backbone of deep neural network is adopt ? how does NMF selection work in Figure 2 ?

Experimental design

Futher experiment are suggested: more than two dataset and more comprehensive comparsion of computation cost.

Validity of the findings

no comment

Additional comments

no comment

Cite this review as

Reviewer 3 ·

Basic reporting

1、The paper is poorly readable and the language needs major revision.

2、The introduction lacks the description of the research significance and application area as well as the structure is not completed, please making major revisions.

3、The references are old in the paper, please replacing the references.

4、The writing of the related work section is logically confusing, please making major revisions.

5、Figure 2 fails to clearly depict the framework proposed in the paper and adding a table of symbol meanings.

6、In section of the method, the MLP module is not described clearly.
7、please Summary part is carefully rewritten.

Experimental design

1、The baseline model selected in this paper is old and lacks the latest benchmark model.

2, In the experimental, please adding ablation experiments, visualization analysis, parameter sensitivity analysis and other experiments.
3、The use of one evaluation criterion in this paper is weakly persuasive, please adding other evaluation criteria.

Validity of the findings

no comment

Additional comments

The following important articles are missing from the references:

1) "Dynamic topic modeling via self-aggregation for short text streams." Peer-to-peer Networking and applications 12 (2019): 1403-1417.
2) "A semantic modeling method for social network short text based on spatial and temporal characteristics." Journal of computational science 28 (2018): 281-293.
3) "A user-based aggregation topic model for understanding user’s preference and intention in social network." Neurocomputing 413 (2020): 1-13.

Cite this review as

---

## Round 0.2 · accepted · Accept

I thank the authors for their efforts to improve the work. The current version can be accepted.

Reviewer 3 ·

Basic reporting

The author has already made revisions based on the comments, and I suggest accepting the publication of this article.

Experimental design

no comment

Validity of the findings

no comment

Additional comments

no comment

Cite this review as